

**Diverse mixing states of amine-containing single particles in Nanjing,**
**China**
Qi En Zhong[1,2], Chunlei Cheng[1,2]*, Zaihua Wang[3]*, Lei Li[1,2], Mei Li[1,2], Dafeng Ge[4,5],
Lei Wang[4,5], Yuanyuan Li[4,5], Wei Nie[4,5], Xuguang Chi[4,5], Aijun Ding[4,5], Suxia
Yang[2,6], Duohong Chen[7], Zhen Zhou[1,2]
[1]Institute of Mass Spectrometry and Atmospheric Environment, Guangdong
Provincial Engineering Research Center for on-line source apportionment system of
air pollution, Jinan University, Guangzhou 510632, China
[2]Guangdong-Hongkong-Macau Joint Laboratory of Collaborative Innovation for
Environmental Quality, Guangzhou 510632, China
[3]Institute of Resources Utilization and Rare Earth Development, Guangdong
Academy of Sciences, Guangzhou 510651, China
[4]Joint International Research Laboratory of Atmospheric and Earth System Sciences
(JirLATEST), School of Atmospheric Sciences, Nanjing University, Nanjing 210023,
China
[5]Collaborative Innovation Center of Climate Change, Jiangsu Province, Nanjing
210023, China
[6]Institute for Environment and Climate Research, Jinan University, Guangzhou
510632, China
[7]State Environmental Protection Key Laboratory of Regional Air Quality Monitoring,
Guangdong Environmental Monitoring Center, Guangzhou 510308, China
*Correspondence to: Chunlei Cheng (chengcl@jnu.edu.cn) and Zaihua Wang (zaihuawang@163.com)
Tel: 86-20-85225991, Fax: 86-20-85225991



**Abstract:** The mixing states of particulate amines with different chemical components are of great significance in studying the formation and evolution processes of amine-containing particles. In this work, the mixing states of single particles containing trimethylamine (TMA) and diethylamine (DEA) are investigated in order to study the formation and aging processes of the single particles using a high-performance single-particle aerosol mass spectrometer located in Nanjing, China, in September 2019. TMA- and DEA-containing particles accounted for 22.8% and 5.5% of the total detected single particles, respectively. The particle count and abundance of the TMA-containing particles in total particles notably increased with enhancement of ambient relative humidity (RH), while the DEA-containing particles showed no increase under a high RH. This result suggested the important role of RH in the formation of particulate TMA. Significant enrichments of secondary organic species, including $^{43}C_2H_3O^+$, $^{26}CN^-$, $^{42}CNO^-$, $^{73}C_3H_5O_2^-$, and $^{89}HC_2O_4^-$, were found in DEA-containing particles, indicating that DEA-containing particles were closely associated with the aging of secondary organics. The particle count and abundance of DEA-containing particles showed a prominent increase during the nighttime, but a sharp decrease during the afternoon. Furthermore, the differential mass spectra of the DEA-containing particles showed a much higher abundance of nitrate during the nighttime than during the daytime. In addition, the number fraction of organic nitrogen species in the DEA-containing particles and ambient NOx both showed consistent increasing trends, similar to the accumulation of DEA-containing particles during the nighttime. This suggested that the nighttime production of particulate DEA might be associated with reactions of gaseous DEA with $HNO_3$ and/or particulate nitrate. Higher abundances of oxalate and glyoxylate were found in DEA-containing particles during the strong photochemistry period when the abundance of DEA-containing particles decreased to the lowest of the total particles. This result suggested a substantial impact of photochemistry on the aging process of DEA-containing particles. Further, greater than 80% of TMA- and DEA-containing particles internally mixed with nitrate, while the abundance of sulfate was higher in the DEA-containing particles (79.3%) than in the TMA-containing particles (55.3%).



In addition, a lesser amount of ammonium was found in the DEA-containing particles
(13.2%) compared with the TMA-containing particles (35%). These observations
suggested that particulate DEA existed both as nitrate and sulfate aminium salts,
while the particulate TMA primarily presented as nitrate aminium salt. Overall, the
different mixing states of the TMA- and DEA-containing particles suggested their
different formation processes and various influencing factors, which are difficult to be
investigated using bulk analysis. These results provide insights into the discriminated
fates of organics during the evolution process in aerosols, which provides a better
illustration of the behavior of secondary organic aerosols.
**Keywords:** Amines; Single particle; Mixing state; Nighttime chemistry; Aminium
salts.

**1 Introduction**
Amines are ubiquitous organic components in aerosols and have a wide range of
sources, including animal husbandry, industrial emissions, vehicle exhaust, biomass
burning, vegetation emissions, and ocean emissions (Ge et al., 2011a; Facchini et al.,
2008; Youn et al., 2015). Due to being highly water-soluble and having strong
alkaline properties, amines play an important role in new particle formation and
substantially contribute to the secondary organic aerosol (SOA) mass (Zhao et al.,
2011; Tao et al., 2016). The formation processes of particulate amines are commonly
associated with the gas-to-particle partitioning of gaseous amines and acid-base
reactions in the particles (Ge et al., 2011b; Pratt et al., 2009). Therefore, ambient
relative humidity (RH) (Rehbein et al., 2011; Zhang et al., 2012), temperature (T)
(Huang et al., 2012), particle acidity (Pratt et al., 2009; Rehbein et al., 2011),
amine-ammonium exchange (Chan and Chan, 2013; Chu and Chan, 2017; Qiu et al.,
2011), and oxidants (Tang et al., 2013; Price et al., 2016) all influence the formation
of particulate amines.
Many field observations have been used to investigate the influence of RH on the
formation of amines. A high RH is beneficial for the formation of amines in most





cases. Zhang et al. (2012) observed a sharp increase in trimethylamine (TMA) during
fog events with high RH. Zhou et al. (2019) found that the concentrations of low
molecular weight (LMW) amines increased significantly under high RH conditions (>
90%). According to the seasonal distributions of amines during the summer and
winter, low T was found to be favorable for the partitioning of gaseous amines into
particles. Huang et al. (2012) found that the number fraction ($N_f$) of amine-containing
particles during winter was four times higher than that during summer.

Gaseous amines can react with sulfuric acid, nitric acid, and organic acids to

form aminium salts, which underscores the important roles of sulfate and nitrate
information of particulate amines (Murphy et al., 2007; Berndt et al., 2010). Berndt et
al. (2010) and Wang et al. (2010) found that the formation of aminium salts via a
neutralization reaction can affect the growth of particles and the generation of SOAs,
which was even stronger than that of $NH_3$. Although amine concentrations are
generally lower than ammonia and ammonium, the amine-ammonium exchange still
contributes to particulate amine formation due to the stronger alkalinity of amines
compared to ammonium (Ge et al., 2011a; Sorooshian et al., 2008). Chan et al. (2013)
found that the exchange reactions between ammonia and amines showed different
reaction rates and product ratios with changes in the aerosol phase state. Qiu et al.
(2011) also found that amines can exchange with ammonium to release ammonia. The
particulate amines produced from the above pathways and reactions constitute a
substantial proportion of the SOAs that impact the physical and chemical properties of
fine particles. In addition to the direct contribution of the SOA mass, the oxidation of
amines by OH radicals, $NO_3$ radicals, and $O_3$ is also a substantial source of SOA
production. Different amines ($NO_3$ radicals, OH radicals, and ozone) exhibit
inconsistent behaviors under the same oxidation environments (Murphy et al., 2007;
Price et al., 2014; Silva et al., 2008). In chamber studies, the oxidation of TMA and
diethylamine (DEA) by OH vs. $NO_3$ radicals resulted in different SOA yields, with
differences greater than one order of magnitude (Tang et al., 2013). Furthermore, even
the same amine showed completely different SOA yields due to OH and $NO_3$ radical
oxidation. Also, the same amine showed distinct trends under the different



temperature changing trends. The formation and oxidation processes of particulate
amines are not well understood, and these processes require additional comprehensive
field observational studies in order to be elucidated.
Most of the field observations did not distinguish between the different behaviors
of each type of amine molecule under the same ambient influencing factors. Actually,
due to the different mixing states of amines with other chemical components, the
amine molecules typically exhibited different behaviors in terms of being oxidized by
OH radicals, forming aminium salts, and altering the hygroscopicity of the particles
(Healy et al., 2015; Cheng et al., 2018; Chu et al., 2015; Price et al., 2016). Therefore,
the formation processes of the different amines are important to reveal the evolution
process of organic aerosols (OAs), and these processes are of great significance to
comprehensively understand the influencing factors of OA production.
Recently, real-time identification of single particles has become an effective
technique to measure the mixing states of diverse amine-containing single particles,
providing a feasible approach to investigate the formation processes of different
particulate amines (Chen et al., 2019; Lian et al., 2020; Cheng et al., 2018). Chen et al.
(2019) found that high RH was favorable for the uptake of DEA, leading to a
DEA-rich substance in the particle phase both during winter and summer. However,
Cheng et al. (2018) and Lian et al. (2020) found that RH was not strongly correlated
with the formation of amine-containing particles during winter and summer. Pratt et al.
(2009) reported that more acidic particles during summer were favorable for the
formation of aminium salts compared with the particles present during autumn,
indicating that the particle acidity affected the gas to particle partitioning of amines.
Rehbein et al. (2011) found more TMA entered the particles as the amount of acidic
particles increased. Based on these studies, although the influences of ambient RH
and particle acidity on the specific type amines formed have been reported, yet
comparative studies between different amines under the same atmospheric
environment using field observation are lacking.
In the present study, the mixing states of TMA- and DEA-containing single
particles are investigated during autumn using a high performance single particle



aerosol mass spectrometer (HP-SPAMS) located in Nanjing, China. Two types of
amine-containing particles exhibited different mixing states with secondarily
produced OA species. The influences of ambient RH, T, and particle acidity on the
mixing states of the two amine-containing particles are evaluated. In addition, the
potential heterogeneous formation of DEA during the nighttime is also discussed. The
results revealed the distinct chemical behaviors of TMA- and DEA-containing
particles and implied the potential role of DEA as an indicator of the aging process of
OA.

## 2 Experimental methods

### 2.1 Sampling site

Ambient single particles were sampled using the HP-SPAMS from September 2–
16, 2019, in Nanjing, China. The sampling site is constantly influenced by
anthropogenic emissions due to its downwind location from urban areas (Figure S1)
(Ding et al., 2013a; Ding et al., 2013b; Ding et al., 2016). The instrument was set up
on top of a small hill (40 m above the ground) on the Nanjing University campus. The
ambient single particles were introduced into the HP-SPAMS through a copper tube.

### 2.2 Instrumentation of the HP-SPAMS

In this work HP-SPAMS (Hexin Analytical Instrument Co., Ltd., China) was
used to detect single particles. The design and principles of SPAMS had previously
been described in detail (Li et al., 2011). In short, particles are introduced into the
aerodynamic lens through a critical orifice at a specific flow rate. Individual particles
are focused and accelerated to specific velocities, which are detected by two
continuous diode Nd:YAG laser beams (532 nm) and then ionized using a pulsed
Nd:YAG laser (266 nm). Finally, the z-shaped bipolar time of the flight mass
spectrometer is used to detect the generated ions. The improvements and
modifications from the SPAMS to the HP-SPAMS are comparatively presented below.
The improvement in the SPAMS primarily includes three portions: the application of
a concentration device, a delay extraction technology, and a multichannel acquisition
technology (Chen et al., 2020; Li et al., 2018). First, the addition of the concentrator



increases the injection flow rate by six times, which allows for improved separation of
gas and particles. Second, since the positions of the ionized ions scatter instead of
being completely linear in the same direction, delayed extraction technology is used
in SPAMS to replace the constant electrical field extraction technique. This delays the
ions in order to obtain sufficient potential energy in the appropriate time under a
pulsed electric field and captures faster ions to improve the resolutions of positive and
negative ions. The mass resolutions of the positive ($> 1000$ at maximum half width)
and negative ($> 2000$ at maximum half width) ion spectra are then significantly
improved. Third, the multichannel acquisition technology is used to divide the signal
into two channels, detecting the high and low intensity signals simultaneously without
signal loss. This new acquisition technology enables a detectable dynamic signal from
5–20000 mV, which is approximately 40 times higher than that of SPAMS.
**2.3 Data analysis**

The size and chemical compositions of single particles obtained using the

HP-SPAMS were analyzed using the Computational Continuation Core (COCO)
toolkit in MATLAB software. According to previous studies that have utilized aerosol
time-of-flight mass spectrometer (ATOFMS) and SPAMS, the amine-containing
particles were identified by querying $^{59}(CH_3)_3N^+$, $^{74}(C_2H_5)_2NH_2^+$, $^{86}(C_2H_5)_2NCH_2^+$,
$^{101}(C_2H_5)_3N^+$, $^{102}(C_3H_7)_2NH_2^+$, and $^{143}(C_3H_7)_3N^+$ (Healy et al., 2015; Angelino et al.,
2001; Cheng et al., 2018; Zhang et al., 2012). In this work, the marker ions of
$^{59}(CH_3)_3N^+$, $^{74}(C_2H_5)_2NH_2^+$, and $^{86}(C_2H_5)_2NCH_2^+$ were detected as the abundant
species, and their particle counts and ratios in the total detected single particles are
shown in Table 1. Single particles containing $^{86}(C_2H_5)_2NCH_2^+$ only accounted for 3.7%
of total particles, which was primarily due to occasional increases on September 5
(Figure S2), possibly due to the outburst of special emissions, such as combustion and
industry. Thus, in this work, particles containing $^{59}(CH_3)_3N^+$ and $^{74}(C_2H_5)_2NH_2^+$ were
selected to discuss the mixing states and formation processes of the particulate amines.
The maker ions of $^{62}NO_3^-$, $^{97}HSO_4^-$, and $^{18}NH_4^+$ were used to identify the nitrate,
sulfate, and ammonium in the amine-containing particles (Zhang et al., 2012). Based
on field and chamber studies using SPAMS and ATOFMS, the $^{43}C_2H_3O^+$ ion was



identified as the representative oxygen-containing organic (Healy et al., 2015; Pratt et
al., 2009). The particles containing $^{26}CN^-$ and $^{42}CNO^-$ were considered to be
representative of the organic nitrogen-containing particles (Pratt et al., 2011). In
addition, the $^{73}C_3H_5O_2^-$ and $^{89}HC_2O_4^-$ ions were designated as glyoxylate and oxalate
markers, respectively (Cheng et al., 2017; Zhang et al., 2020).

## 3 Results and discussion

### 3.1 Characteristics of amine-containing particles

In this work, the amine-containing particles accounted for 32.1% of total

detected single particles, which was higher than in previously reported results for the
Pearl River Delta (PRD) region (9.4%−11.1%) and Chongqing (8.3%−12.7%), China.
The TMA-containing particles showed a much higher abundance in the total particles
(22.8%) than the DEA-containing particles (5.5%) (Table 1), which could have been
due to their differential emissions and atmospheric processing (Cheng et al., 2018;
Chen et al., 2019; Liu et al., 2020; Ge et al., 2011a). Temporal variations in
meteorological parameters, $PM_{2.5}$ concentration, and the count of amine-containing
particles are shown in Figure 1. Although the TMA- and DEA-containing particles
exhibited similar temporal trends at a lower particle count, their increasing peaks
appeared at different periods, suggesting that the reasons for their increase in the
particle count were different. Generally, peaks in the DEA-containing particles
frequently appeared during the nighttime, which was possibly due to their enhanced
source emissions and/or favorable nighttime production (Tang et al., 2013). The
ambient RH was relatively high during the entire sampling period (74 ± 14%),
especially from September 5–7, when the count of TMA-containing particles sharply
increased. However, no obvious enhancement in DEA-containing particles count was
found, which suggested other influencing factors on their formation process in
addition to the ambient RH.

Additionally, the periods of high concentration of the amine-containing particles

were not consistent with the increase in $PM_{2.5}$ concentration, which could have been
due to the integrated effects of the emission sources and the secondary formation



processes. The backward trajectories of the air masses (48 h, 500 m) associated with
the spatial distributions of the two amine-containing particles during the entire
sampling period are presented in Figure 2. More than 70% of the air masses (Clusters
1 and 4) were from east of the sampling site, which were both connected with
anthropogenic emissions in the Yangtze River Delta (YRD) and marine sources in the
East China Sea. TMA-containing particles were primarily from the air masses of
Cluster 1 and Cluster 4, while the DEA-containing particles were associated with the
air masses of Cluster 3 and Cluster 4, which underwent long-range transport. These
results suggested potential different emission sources and atmospheric formation
processes of TMA- and DEA-containing particles, which was further investigated by
examining their mixing states.

Diurnal variations of TMA- and DEA-containing particles are shown in Figure 3.

The particle count of TMA-containing particles and their abundance in the total
particles exhibited identical variation patterns. These patterns exhibited significant
increases in the morning, possibly associated with direct emissions from vehicle
exhaust (Chen et al., 2019; Cheng et al., 2018). The DEA-containing particles showed
a completely different diurnal pattern compared with the TMA-containing particles.
DEA-containing particles increased during nighttime, but sharply decreased during
the afternoon, when the photochemistry was the most active. The nighttime increase
could have been due to the high ambient RH and/or enhanced heterogeneous reactions
(Zhou et al., 2019; Huang et al., 2012; Zhang et al., 2019). However, since the
increase in the DEA-containing particles was not prominent under high RH (Figure 1),
enhanced heterogeneous production of particulate DEA could be a more reasonable
explanation. The decrease in the DEA-containing particles during the afternoon could
have been associated with the photodegradation of DEA and/or repartitioning of
particulate DEA under high temperatures during the day (Murphy et al., 2007; Ge et
al., 2011b; Pitts et al., 1978). In order to investigate the impact of ambient RH on the
formation process of amine-containing particles, the particle counts of amine particles
and the relative peak areas (RPAs) of amines in the particles with an increase in the
ambient RH are presented in Figure 4. The particle count of TMA-containing particles



and the RPA of the TMA showed remarkable increasing trends with an enhancement
in RH during the entire sampling period. This result suggested a significant role of
RH in the formation of particulate TMA. This was consistent with a field study
conducted in Guangzhou, China, which also found an instant increase in
TMA-containing particles after the occurrence of fog events (Zhang et al., 2012). In
contrast, the particle count of DEA-containing particles only exhibited increased RH
range between 70–80% and decreased with the continuous increase in the RH.
Additionally, the RPA of the DEA showed little change with an increase in the RH,
which suggested the minor influence of a change in RH on the formation of
particulate DEA. The different responses of TMA and DEA with RH changes
signified their different formation processes in the particles.
**3.2 Different mixing states of amine-containing particles**

It is important to understand the chemical compositions of amine-containing

particles in order to understand their mixing states and track their formation processes.
Hence, the positive and negative mass spectra of the amine-containing particles are
shown in Figure 5. Generally, TMA- and DEA-containing particles both contained
amine marker ions, as well as organic fragments such as $^{27}C_2H_3^+$, $^{37}C_3H^+$, $^{43}C_2H_3O^+$,
$^{51}C_4H_3^+$, and $^{61}C_5H^+$ in the positive mass spectra. In addition, their negative mass
spectra were both characterized by nitrate, sulfate, and nitric acid ($^{125}H(NO_3)_2^-$).
However, DEA-containing particles contained many more organic fragments and a
higher abundance of hydrocarbon clusters than the TMA-containing particles. In the
positive mass spectra, the abundance of the hydrocarbon fragments with an m/z below
60 was 2–3 times higher in DEA-containing particles than that in TMA-containing
particles. In addition, hydrocarbon fragments with an m/z above 60 were barely
detectable in TMA-containing particles, while abundant hydrocarbon fragments with
an m/z ranging from 60–150 were observed in DEA-containing particles. Furthermore,
the DEA-containing particles also contained abundant secondary organic marker ions,
including organic nitrogen ($^{26}CN^-$ and $^{42}CNO^-$), acetate ($^{59}C_2H_3O_2^-$), glyoxylate
($^{73}C_3H_5O_2^-$), and oxalate ($^{89}HC_2O_4^-$) in the negative mass spectra, and these were not
found in the TMA-containing particles. This was in accordance with the linear





regressions between these secondary organic ions containing particles with two
amine-containing particles (Table 2), which showed no correlations in the
TMA-containing particles ($r^2 < 0.1$), but good correlations in the DEA-containing
particles ($r^2 > 0.57$). The differential mass spectral features in the distributions of
organics in the two amine-containing particles (Figure 6) suggested that more
secondary organics accumulated in DEA-containing particles than in TMA-containing
particles. This result also implied that multiple factors influenced the mixing state of
DEA-containing particles in addition to ambient RH.

In order to further characterize the mixing states of DEA-containing particles

with secondary organic ions, temporal variations and diurnal patterns of secondary
organic ions in the DEA-containing particles are presented in Figure 7. As the
oxidation products of various organics, the abundances of glyoxylate and oxalate
commonly increased between 12:00 and 18:00 (Figure 7), when the photochemistry
was most active during the daytime. This result suggested the deep photochemical
aging state of DEA-containing particles. This might explain the decrease in the
particle counts of DEA-containing particles (Figure 3), which was partially associated
with the photo-degradation of particulate DEA. Pitts et al. (1978) reported that under
sunlight particulate DEA decomposed to acetamide, while DEA in the gas phase was
oxidized to acetaldehyde, PAN, amide, and imine. Gaseous DEA can be oxidized into
carbonyl compounds and other amines by ozone and OH radicals that primarily
include acetaldehyde and N-ethylethanimine (Tuazon et al., 2011; Tong et al., 2020).
The organic nitrogen markers of $^{26}CN^-$ and $^{42}CNO^-$ showed different temporal trends
with glyoxylate and oxalate. Although the abundances of organic nitrogen markers
also increased after 12:00 like oxalate, the markers still maintained high abundances
during the nighttime, when glyoxylate and oxalate sharply decreased. This result
suggested that the aging process of organics during the nighttime was slower than that
during the afternoon in the DEA-containing particles. Thus, the increase of
DEA-containing particles (Figure 3) could have been due to the enhanced production
of particulate DEA during the nighttime. In addition, the differential mass spectra of
DEA-containing particles (Figure 8) between the nighttime (22:00–02:00) and


daytime (14:00–18:00) showed a significant enrichment of nitrate during the
nighttime. This result suggested that nighttime production of particulate DEA was
associated with gaseous HNO$_3$ and/or particulate nitrate (Price et al., 2016).

Temporal variations in NOx and the N$_f$ of the DEA-containing particles in the

total detected particles are presented in Figure 9. They showed similar increasing
patterns during the nighttime, and a high abundance of nitrate in the DEA-containing
particles was also observed. This result suggested the important role of nitrate in the
formation of particulate DEA. The particulate DEA during the nighttime could have
been produced from the reaction of gaseous DEA with HNO$_3$ during the gas phase
(R1) followed by the gas to particle partitioning (R2) and/or the direct heterogeneous
formation pathway (R3) (Price et al., 2016; Nielsen et al., 2012). The high ambient
concentration of NOx is favorable for the production of NO$_3$ radicals and the
heterogeneous production of nitrate, which might explain the distinct enhancement of
the DEA-containing particles. However, the same formation pathways were also
applied to TMA, yet there was no significant increase in the N$_f$ of the
TMA-containing particles in the total particles (Figure 3). This could have been due to
the different particle/gas dissociation constant ($K_p$) for DEA·HNO$_3$ and TMA·HNO$_3$,
which was several orders of magnitude lower than that for DEA·HNO$_3$ (7.01E-09)
compared with TMA·HNO$_3$ (1.65E-06) at 25 ℃ (Price et al., 2016; Ge et al., 2011b).
During the entire sampling period, the ambient temperature during the nighttime was
approximately 24 ℃. Thus, the produced DEA·HNO$_3$ tended to stay in the particles,
while a portion of the TMA·HNO$_3$ repartitioned back into the gas phase. This resulted
in an insignificant increase in the TMA-containing particles. Further studies should
consider the influence of the different volatilities of DEA·HNO$_3$ and TMA·HNO$_3$ on
the formation of particulate amines in chamber experiments due to the lack of
quantitative results in this study.

$(CH_3CH_2)_2NH_{(g)} + HNO_{3(g)} \rightarrow (CH_3CH_2)_2N \cdot HNO_{3(g)}$.                    (R1)

$(CH_3CH_2)_2N \cdot HNO_{3(g)} \leftrightarrow (CH_3CH_2)_2N \cdot HNO_{3(s)}$.                    (R2)

$(CH_3CH_2)_2NH_{(g)} + NO_3^-{}_{(s)} \rightarrow (CH_3CH_2)_2N \cdot HNO_{3(s)}$.                    (R3)

**3.3 Formation of aminium salts**



To study the acid-base reactions of TMA and DEA with sulfate and nitrate, the
$N_f$s of nitrate-, sulfate-, and ammonium-containing particles in total detected particles
and amine-containing particles are shown in Table 3. More than 80% of TMA- and
DEA-containing particles internally mixed with nitrate, which was higher than the $N_f$
of nitrate in the total particles (72%). Interestingly, the $N_f$ of sulfate in
DEA-containing particles (79.3%) was much higher than that in TMA-containing
particles (55.3%) and in the total particles (60.1%). This was similar to a study
performed by Lian et al. (2020) that found a stronger correlation between
$^{86}(C_2H_5)_2NCH_2^+$ with sulfate than that between TMA and sulfate. In addition, in this
work, robust linear correlations ($r^2 > 0.9$) between nitrate-containing particles and
amine particles were both observed in TMA- and DEA-containing particles (Table 2).
However, a weak linear correlation ($r^2 = 0.32$) was found between the
sulfate-containing particles and the TMA-containing particles, while a better linear
correlation ($r^2 = 0.86$) was observed in the DEA-containing particles. According to
reported studies, the vapor pressure of diethylaminium sulfate (DEAS) ($0.2*10^{-12}$–
$12.8*10^{-12}$ Pa) was three orders of magnitude lower than that of trimethylaminium
sulfate (TMAS) ($0.6*10^{-9}$–$1.8*10^{-9}$ Pa) at 298 k. In addition, the enthalpy of
evaporation was higher than that of TMAS (DEAS: $168 \pm 5$ kJ mol$^{-1}$; TMAS: $114 \pm 2$
kJ mol$^{-1}$) (Lavi et al., 2013). Therefore, the thermo-stability of DEAS was stronger
than TMAS (Qiu and Zhang, 2012), which led to the higher $N_f$ of sulfate in the
DEA-containing particles than in the TMA-containing particles.
The $N_f$ of ammonium in DEA-containing particles (13.2%) was lower than in
TMA-containing particles (35%) and total particles (19.4%). The low abundance of
$NH_4^+$ in DMA-containing particles had been observed in our previous studies in the
PRD region (Cheng et al., 2018), which was partially attributed to the
ammonium-amine exchange reactions in the particles. The related laboratory
experiments primarily involved the preferential uptake of LMW amines in the $H_2SO_4$
particles (Sauerwein and Chan, 2017; Chan and Chan, 2013; Chu and Chan, 2017). In
this work, the distinct low $N_f$ of $NH_4^+$ in the DEA particles suggested the possible
displacement of $NH_4^+$ by DEA. Moreover, the higher abundance of sulfate in DEA


388 particles than in TMA particles was more favorable for the occurrence of

389 ammonium-amine exchange reactions in DEA particles. This disparity could imply

390 differential roles of DEA and TMA in the new particle formation process (Wang et al.,

391 2010; Yin et al., 2011; Zhao et al., 2011).

392  The temporal trends of the $N_f$s of nitrate-, sulfate-, and ammonium-containing

393 particles in TMA and DEA particles are shown in Figure 10. The $N_f$ of

394 nitrate-containing amine particles exhibited similar variation patterns with each type

395 of amine particle, while the $N_f$ of sulfate-containing amine particles only showed a

396 similar variation pattern with DEA-containing particles. Although aminium nitrate

397 and sulfate salts were both produced in TMA- and DEA-containing particles, the

398 different temporal trends of sulfate and nitrate in the two amine particles suggested

399 that both sulfate and nitrate DEA salts existed in the DEA-containing particles, while

400 nitrate TMA salt dominated in TMA-containing particles (Cheng et al., 2018; Pratt et

401 al., 2009). This difference in the form of aminium salts could signify the potential

402 different influences in the hygroscopic property of secondarily processed particles

403 internally mixing with different amines (Rovelli et al., 2017; Clegg et al., 2013; Lavi

404 et al., 2013). The relative acidity ratio (Ra), defined as the ratio of the sum of the

405 sulfate and nitrate peak areas to the ammonium peak area, has been proposed in field

406 studies that use single particle mass spectrometry to roughly estimate particle acidity

407 (Huang et al., 2013; Cheng et al., 2018). In this work, temporal variations in Ra in

408 TMA-containing particles ($Ra_1$) and DEA-containing particles ($Ra_2$) are shown in

409 Figure 11. The average $Ra_1$ was $6.3 \pm 1.8$ in TMA-containing particles, and $Ra_2$ was

410 $36.1 \pm 21.8$ in DEA-containing particles. This result could suggest a more acidic

411 nature of DEA particles than TMA particles. However, after including the peak area

412 of amines (TMA/DEA) in the calculation of Ra, the new Ra′ was reduced to $2.1 \pm 0.5$

413 in the TMA-containing particles and $6.5 \pm 1.2$ in the DEA-containing particles. The

414 gap of Ra between the two amine particles significantly decreased after including

415 amines in the calculation. The larger reduction ratio of Ra′ in DEA-containing

416 particles than in TMA-containing particles suggested the effective buffering effect of

417 amines under the absence of ammonium in the particles.





**3.4 Implications of the diverse mixing states of amines particles**


The mixing states and formation processes of the two amine-containing particles
were investigated under the same atmospheric environment, and their different
atmospheric behaviors against the same influencing factors suggested their
differential contributions to SOA mass. The prominent impact of ambient RH on the
formation of particulate TMA suggested a significant role for gas-particle partitioning
process to the high water-soluble species in the SOA. However, the slight influence of
RH on the formation of the particulate DEA implied the inconsistent role of high RH
on the same group of water-soluble organic molecules. In addition, the distinct
distribution patterns of secondary organic species in two amine-containing particles
also signified that the mixing states of the OA are important to explore their formation
processes. Furthermore, the heterogeneous processing of the DEA-containing
particles during the nighttime and the photochemical degradation of the DEA during
the daytime both generated more fractions of nitrogen- and oxygen-containing species
in the particles than in the TMA-containing particles. This result suggested different
roles of particulate TMA and DEA in the evolution of hygroscopicity and aging state
of the SOA. In summary, understanding mixing states and formation processes of
different amines in single particles is of great significance to reveal the unique
response of each type of amine to the same atmospheric environment. Single-particle
analysis provided insights into the mixing states of specific organic species to further
understand the formation process of the SOA.
**4 Summary and conclusions**
TMA- and DEA-containing single particles were collected and analyzed on
September 2019 using HP-SPAMS in Nanjing, China, and accounted for 22.8% and
5.5% of total detected particles, respectively. The mixing states and formation
processes of TMA- and DEA-containing particles were studied. With increased RH,
the counts of particulate TMA and the RPA of the TMA displayed an obvious upward
trend, while the particle count of the particulate DEA slightly increased when the RH
was 70–80%. In addition, the RPA of the DEA showed no difference in reaction to



RH change during the entire sampling period. This suggested a differential role for
ambient RH during the formation processes of particulate TMA and DEA. The
possible formation processes were further evaluated by analyzing the mixing states of
the amine-containing particles. The mass spectra of the amine-containing particles
showed that the secondary organic species were enriched in the DEA-containing
particles. The differential distributions of the secondary ions effectively explained the
sharp increase in DEA-containing particles during the nighttime, which could have
been due to the heterogeneous reactions of gaseous DEA with $HNO_3$ and/or nitrate
particles. The prominent decrease in the DEA-containing particles during the
afternoon was attributed to photo-degradation of particulate DEA. Due to the
differences in the thermodynamic properties, the $N_f$ of sulfate in the particulate DEA
was higher than that in the particulate TMA. The amine-ammonium exchange
reaction resulted in particulate DEA containing less $NH_4^+$. In addition, the particulate
DEA was abundant in sulfate, which was more favorable for the exchange of amine
and ammonium. The higher relative acidity ratio in DEA-containing particles relative
to TMA-containing particles could suggest that DEA particles are more acidic. After
including the peak area of amines (TMA/DEA) in the calculation, the larger reduction
ratio of the Ra′ in DEA-containing particles than in TMA-containing particles
suggested the effective buffering effect of amines under the absence of ammonium in
the particles. These results revealed the distinct mixing states and chemical behaviors
of TMA- and DEA-containing single particles and could imply a potential role for
DEA as an indicator of the OA aging process.

**Data availability**

The observational data, including HP-SPAMS and the meteorological parameters,

obtained in this study are available from the corresponding authors upon request
(chengcl@jnu.edu.cn).

**Author contribution**


**Qi En Zhong, Chunlei Cheng, Zaihua Wang:** methodology, writing original
draft. **Dafeng Ge, Lei Wang, Yuanyuan Li, Wei Nie, Xuguang Chi, Aijun Ding:**
methodology, sampling. **Lei Li, Mei Li, Suxia Yang, Duohong Chen, Zhen Zhou:**
providing discussions and helping to revise original draft.

**Competing interests**
I declare that I or my co-authors have competing interests as follows: Aijun Ding
is editor of ACP.

**Acknowledgements**: This work was financially supported by the National Key
Research and Development Program of China (Grant No. 2018 YFE0106900), the
National Natural Science Foundation of China (Grant Nos. 41805093, 41827804 and
41875175), the NSFC of Guangdong Province (Grant No. 2021A1515011206), the
Guangzhou Economic and Technological Development District International Science
and Technology Cooperation Project (Grant No. 2018GH08), the National Research
Program for Key Issues in Air Pollution Control (Grant No. DQGG0107), the Pearl
River Nova Program of Guangzhou (No. 201806010064), and GDAS' Project of
Science and Technology Development (2021GDASYL-20210103058).

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


**Tables and figures**

**Table list:**
**Table 1.** Summary of the major species of detected amine-containing particles and
fragments in September in Nanjing, China.
**Table 2.** The linear correlations ($r^2$) between secondary ion-containing amine
particles within TMA- and DEA-containing particles.
**Table 3.** Number fractions sulfate, nitrate, and ammonium in TMA-containing
particles, DEA-containing particles, and total particles.

**Figure caption:**
**Figure 1.** Temporal variations in relative humidity (RH), temperature (T), $O_3$
concentration, $PM_{2.5}$ concentration, wind speed, wind direction, and TMA- and
DEA-containing particles during the entire sampling period.
**Figure 2.** Backward trajectories (48 h) of air masses at 500 m above the ground
during the sampling period: (a) TMA-containing particles counts; (b) DEA-containing
particle counts. C1 to C4 represent cluster 1 to cluster 4.
**Figure 3.** The diurnal variations in particle counts and number fractions of the two
amine-containing particles in total particles during the entire sampling period.
**Figure 4.** Particle counts of amine-containing particles and relative peak area (RPA)
of the two amines in single particles, with an increase in ambient RH. (a, c)
TMA-containing particles; (b, d) DEA-containing particles.
**Figure 5.** Mass spectra of TMA- and DEA-containing particles during the entire
sampling period.
**Figure 6.** Differential mass spectra between DEA- and TMA-containing particles.
**Figure 7. (a)** Temporal trends of the relative peak areas (RPAs) of $^{73}C_3H_5O_2^-$,
$^{89}HC_2O_4^-$, $^{26}CN^-$, and $^{42}CNO^-$ in DEA-containing particles. **(b)** Diurnal variations in
the relative RPAs of $^{73}C_3H_5O_2^-$, $^{89}HC_2O_4^-$, $^{26}CN^-$, and $^{42}CNO^-$ in DEA-containing
particles.





**Figure 8.** Differential mass spectra of DEA-containing particles between 22:00–02:00
and 14:00–18:00.
**Figure 9.** Temporal trends of RPA nitrate in DEA-containing particles, number
fraction of DEA-containing particles in total particles, and NOx concentration.
**Figure 10.** Temporal trends of TMA- and DEA-containing particle counts, and
number fractions of nitrate, sulfate, and ammonium in TMA- and DEA-containing
particles.
**Figure 11.** Temporal trends of the relative acidity ratios (Ra, Ra') in TMA- and
DEA-containing particles.



**Tables**
**Table 1.** Summary of the major species of detected amine-containing particles and
fragments in September in Nanjing, China.

| Alkylamine assignment | Count | Percentage (%) |
|---|---|---|
| All detected particles | 4 693 931 | |
| $^{59}(CH_3)_3N^+$ (TMA)-containing particles | 1072143 | 22.8 |
| $^{74}(C_2H_5)_2NH_2^+$ (DEA) -containing particles | 259913 | 5.5 |
| $^{86}(C_2H_5)_2NCH_2^+$ (TEA)-containing particles | 172621 | 3.7 |


**Table 2.** The linear correlations ($r^2$) between secondary ion-containing amine
particles within TMA- and DEA-containing particles.

| | TMA particles | DEA particles |
|---|---|---|
| $^{26}CN^-$ | 0.13 | 0.70 |
| $^{42}CNO^-$ | 0.09 | 0.70 |
| $^{73}C_3H_5O_2^-$ | 0.01 | 0.66 |
| $^{89}HC_2O_4^-$ | 0.09 | 0.57 |
| $^{43}C_2H_3O^+$ | 0.05 | 0.90 |
| $^{62}NO_3^-$ | 0.93 | 0.90 |
| $^{97}HSO_4^-$ | 0.32 | 0.86 |
| $^{18}NH_4^+$ | 0.50 | 0.28 |


**Table 3.** Number fractions sulfate, nitrate, and ammonium in TMA-containing
particles, DEA-containing particles, and total particles.

| | TMA particles | DEA particles | Total particles |
|---|---|---|---|
| Sulfate | 55.3 | 79.3 | 60.1 |
| Nitrate | 81.6 | 81.8 | 72.0 |
| Ammonium | 35.0 | 13.2 | 19.4 |


## Figures

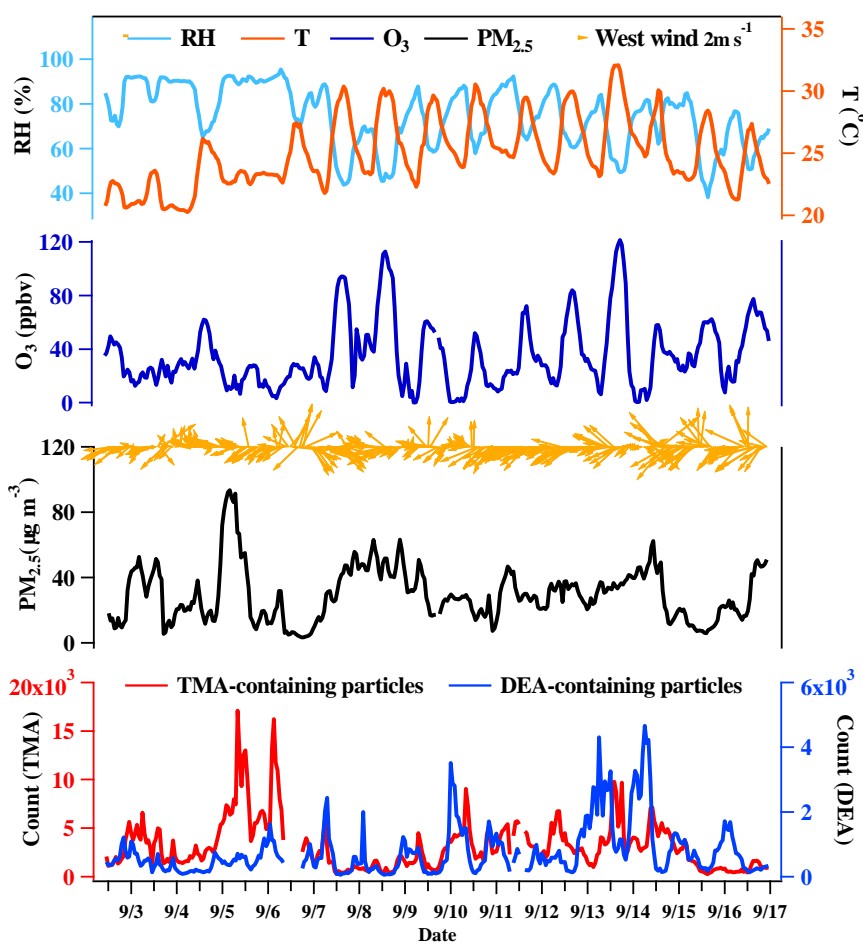

**Figure 1.** Temporal variations in relative humidity (RH), temperature (T), $O_3$ concentration, $PM_{2.5}$ concentration, wind speed, wind direction, and TMA- and DEA-containing particles during the entire sampling period.





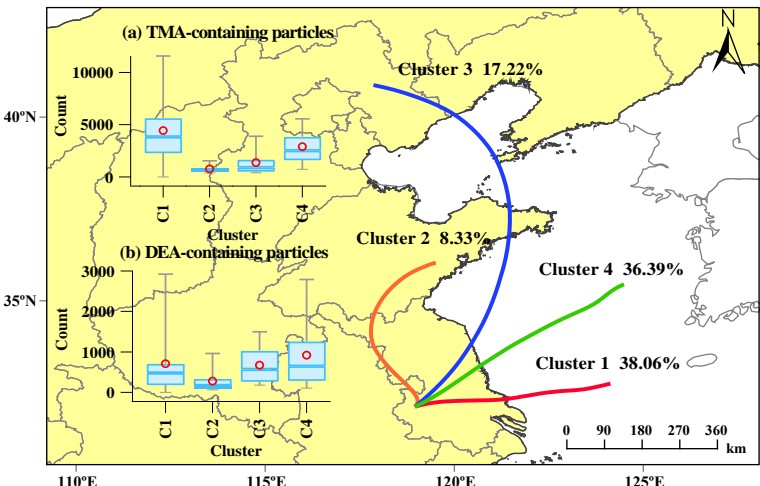


**Figure 2.** Backward trajectories (48 h) of air masses at 500 m above the ground during the sampling period: (a) TMA-containing particles counts; (b) DEA-containing particle counts. C1 to C4 represent cluster 1 to cluster 4.



















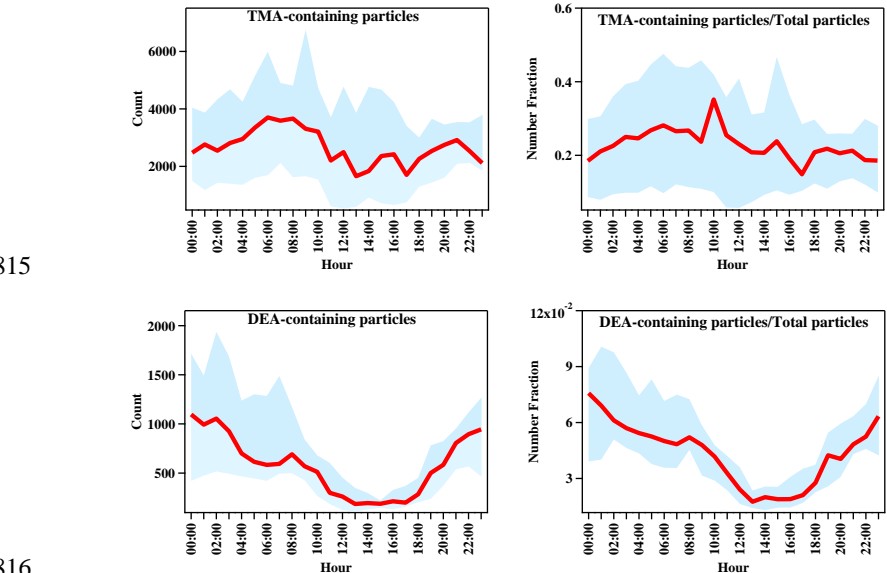

**Figure 3.** The diurnal variations in particle counts and number fractions of the two amine-containing particles in total particles during the entire sampling period. The shaded areas represent the 75[th] and 25[th] percentiles.



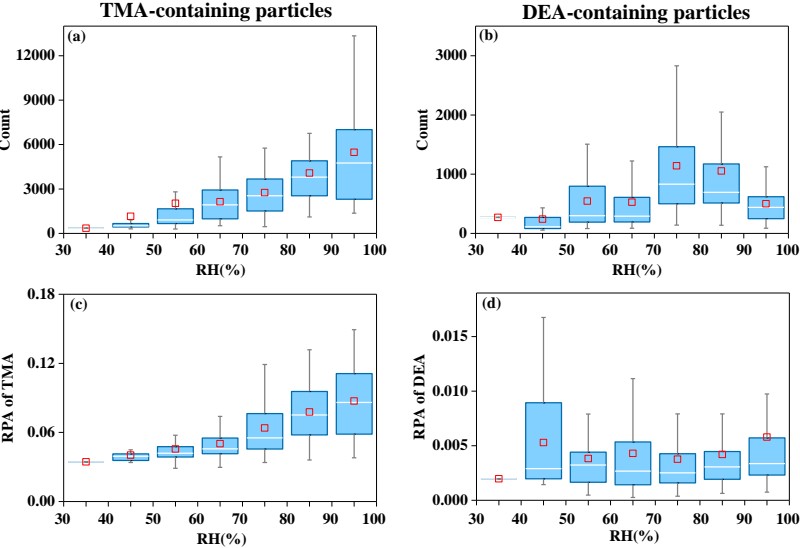

**Figure 4.** Particle counts of amine-containing particles and relative peak area (RPA) of the two amines in single particles, with an increase in ambient RH. (a, c) TMA-containing particles; (b, d) DEA-containing particles. Squares represent the average values. The line inside the box indicates the median. Upper and lower boundaries of the box represent the 75th and the 25th percentiles; the whiskers above and below each box represent the 95th and 5th percentiles.





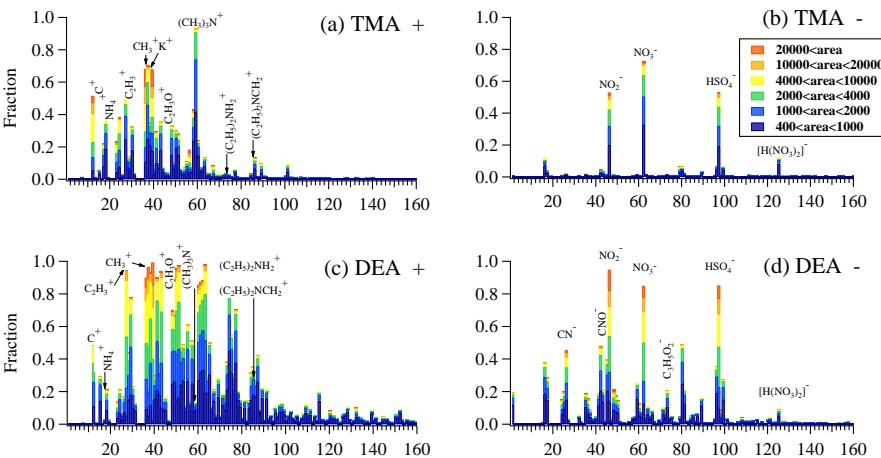

**Figure 5.** Mass spectra of TMA- and DEA-containing particles during the entire sampling period.

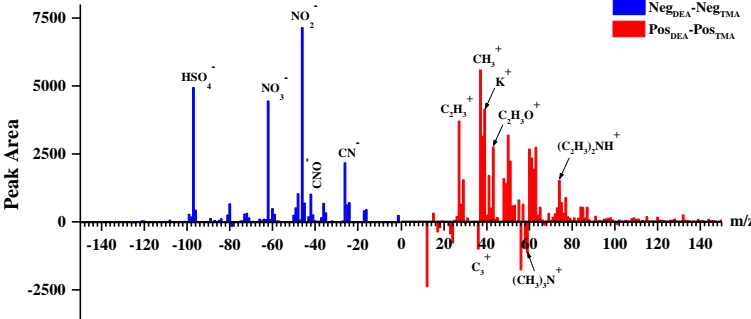

**Figure 6.** Differential mass spectra between DEA- and TMA-containing particles.

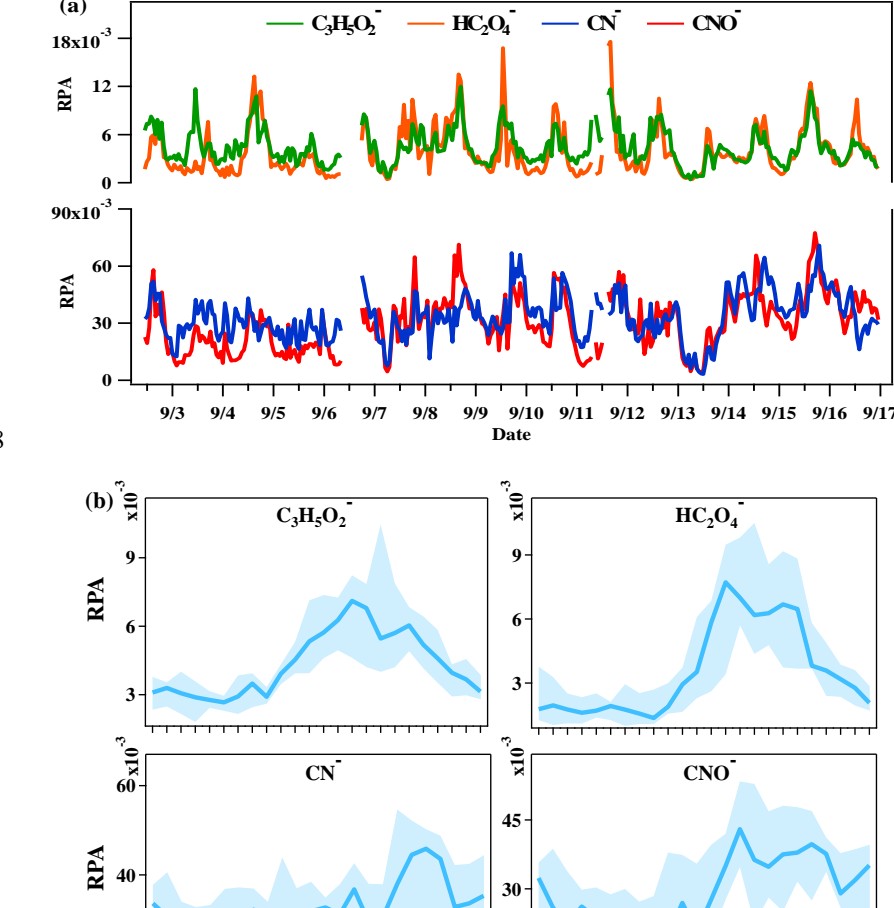

868

869

**Figure 7. (a)** Temporal trends of the relative peak areas (RPAs) of $^{73}C_3H_5O_2^-$, $^{89}HC_2O_4^-$, $^{26}CN^-$, and $^{42}CNO^-$ in DEA-containing particles. **(b)** Diurnal variations in the relative RPAs of $^{73}C_3H_5O_2^-$, $^{89}HC_2O_4^-$, $^{26}CN^-$, and $^{42}CNO^-$ in DEA-containing particles. The shaded areas represent the 75th and 25th percentiles.








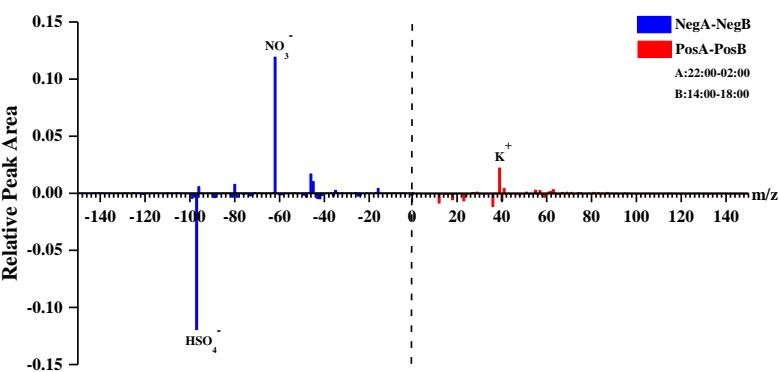


**Figure 8.** Differential mass spectra of DEA-containing particles between 22:00–02:00
and 14:00–18:00.





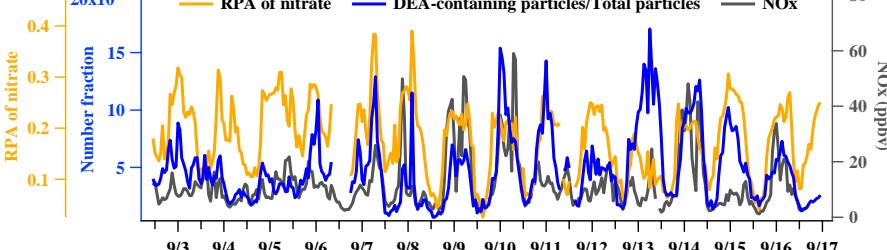


**Figure 9.** Temporal trends of RPA nitrate in DEA-containing particles, number
fraction of DEA-containing particles in total particles, and NOx concentration.





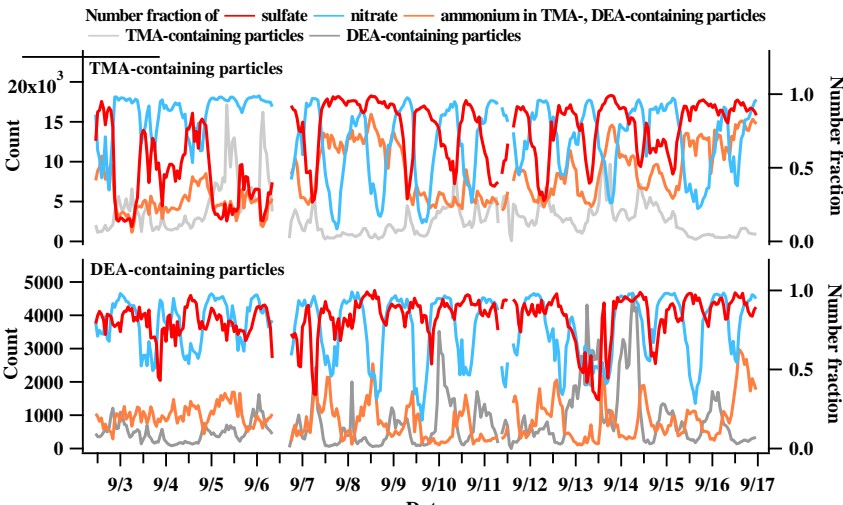

**Figure 10.** Temporal trends of TMA- and DEA-containing particle counts, and number fractions of nitrate, sulfate, and ammonium in TMA- and DEA-containing particles.

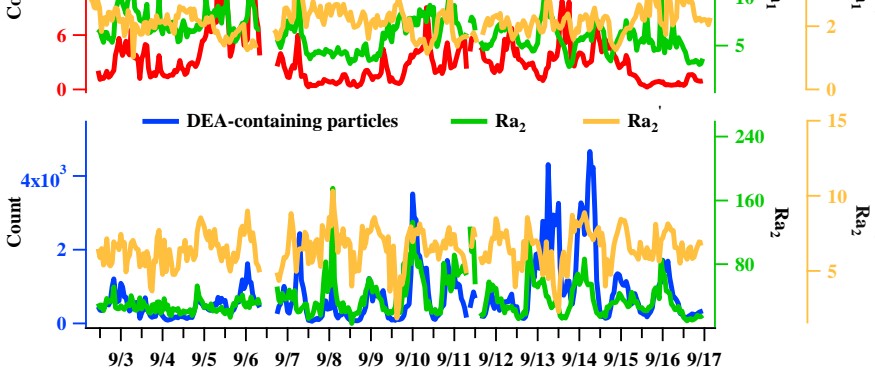

**Figure 11.** Temporal trends of the relative acidity ratios (Ra, Ra$^{'}$) in TMA- and DEA-containing particles.