# Peer review of "Diverse mixing states of amine-containing single particles in Nanjing,"

_Atmospheric Chemistry and Physics, 2021_

## Author Comment (AC1)

**Response to the comments of Anonymous Referee #1**

[Atmospheric Chemistry and Physics, MS ID: acp-2021-593]
Title: Diverse mixing states of amine-containing single particles in Nanjing, China

**General comments:**
This paper reported the different mixing states of two amine-containing single particles, and discussed their different formation processes and influencing factors. The results showed the ambient RH was the major reason for the increase of TMA-containing particles, while the formation and influencing factors for DEA-containing particles were more complicated due to the enrichment of secondary organics. The accumulation of nitrate and organic nitrogen species in DEA-containing particles were explained as the nighttime production of particulate DEA via reactions with gaseous $HNO_3$ and/or particulate nitrate. The different mixing states of TMA and DEA with sulfate, nitrate and ammonium were investigated, which implied that particulate DEA existed both as nitrate and sulfate aminium salts, while the particulate TMA primarily presented as nitrate aminium salt.

Since most of the field observations did not distinguish between the different behaviors of each type of amine molecule under the same ambient influencing factors, these results and conclusions based on single particle analysis are quite interesting, which shows the different fates and formation processes of two particulate amines. Given the substantial roles of amines in new particle formation and contributions to the SOA mass, the distinct mixing states and influencing factors of amine-containing particles are of great significance to reveal the formation and evolution processes of bulk OA. Overall, this manuscript is well written and results are interesting with novelty. I recommend this manuscript should be accepted for publication with one substantial revision.

**Response**: Thank you for your comments and contribution to our manuscript. We have revised the whole manuscript according to your comments, and have prepared the point-by-point responses to the comments. These valuable comments are of great help to the revision and improvement. Our responses to the comments are listed below and the correspondent revisions are highlighted in blue color in the revised manuscript.

Anything about our paper, please feel free to contact me at chengcl@jnu.edu.cn

Best regards!
Sincerely yours

Chunlei Cheng
October 11, 2021

**Specific comments and point by point responses:**

**Major comments:**

1.  In the introduction part.

I asked the authors consider why your method is suitable to do this work. For mixing state of individual particles, some scientists used the electron microscope to observe very good mixing state of inorganic and organic. However, there is no ability to obtain the amine species in organic compounds. Then the SPAMS not only can get the organic compounds of single organic particles, but they can specifically distinguish the amines. Therefore, the authors can find some related studies for this part.

    **Response:** Indeed, the mixing states of organics in single particles are commonly investigated by electron microscope and mass spectrometry (Li et al., 2016a; Cheng et al., 2018; Yu et al., 2019). The technique of electron microscope coupled with energy dispersive X-ray spectrometry (EDX) can provide the morphological, physical and some chemical information of single particles, and also identify the particle phases of some crystalline particles (Li et al., 2016b; Li et al., 2021). However, electron microscope coupled with EDX cannot distinguish and identify the specific organic molecules (Li et al., 2016b), which is incapable of analyzing amines in single particles. Despite the disadvantages in determining the morphological and physical information of single particles, the technique of mass spectrometry (mainly including ATOFMS and SPAMS) can identify the real-time presence and relative abundance of specific organic ions, which provides substantial data to understand the sources, formation and evolution processes of selective organic markers (Cheng et al., 2018; Chen et al., 2019; Angelino et al., 2001). In this work the different mixing states of two amines were clearly identified via HP-SPAMS, which confirmed the feasibility of single particle mass spectrometry in determining the mixing states of amines.

    The related discussions about electron microscope and mass spectrometry have been added in the revised manuscript. "The mixing states of organics in single particles are commonly investigated by electron microscope and mass spectrometry (Yu et al., 2019; Li et al., 2016a; Cheng et al., 2018). The technique of electron microscope coupled with energy dispersive X-ray spectrometry (EDX) can provide the morphological, physical and some chemical information of organics in single particles (Li et al., 2016b; Li et al., 2021). However, electron microscope coupled with EDX cannot distinguish and identify the specific organic molecules (Li et al., 2016b), which is incapable of analyzing amines in single particles. The technique of single particle mass spectrometry can identify the real-time presence and relative abundance of specific organic ions, which provides substantial data to understand the sources, formation and evolution processes of selective organic markers, providing a feasible approach to investigate the formation processes of different particulate amines (Cheng et al., 2018; Chen et al., 2019; Angelino et al., 2001)." has been added in the lines 125-137.

2.   Here I also feel that the authors need to tell us why you select the Nanjing city as the sampling site. In the introduction part, you might tell us what the sampling site is suitable. Then you conducted the field campaign. Seemly, the authors did not give us strong reason for this point.

   **Response:** Thanks for your suggestion. We have added the description of importance of sampling campaign in Nanjing as follows:
   "Nanjing is a typical megacity in the Yangtze River Delta (YRD), which is downwind of other megacities including Shanghai, Changzhou, Suzhou, and Wuxi (Figure S1). In addition, Nanjing suffers from heavy loadings of anthropogenic pollutants as well as the complex impacts of biogenic and ship emissions (Xu et al., 2021; Zhao et al., 2020; Ding et al., 2013a). The investigation of mixing states of amines in Nanjing helps to explorer the formation and evolution processes of OAs." has been added in the lines 151-157.
   "The sampling site is constantly influenced by anthropogenic emissions due to its downwind location from urban areas (Figure S1)" has been revised to "The campaign was conducted at the Station for Observing Regional Processes of the Earth System (SORPES) station in Nanjing University Xianlin Campus (Ding et al., 2016; Ding et al., 2013a; Ding et al., 2013b; Liu et al., 2021)." in the lines 168-171.

3.   In the method part, I am interesting with the cut off size of aerosol particles in the HP-SPAMS because the study work is related with the new particle formation. I didn't notice what size range of particles were analyzed in the SPAMS.

   **Response:** Generally, the particle size obtained by SPAMS and HP-SPAMS ranges from 0.2 to 2.0 μm (Li et al., 2018; Li et al., 2011). The typical size distributions of ambient single particles detected by SPAMS are presented in the following Figure R1. The detection efficiency of SPAMS and HP-SPAMS varies at different size diameter due to matrix effect and transmission efficiency. Thus, the field observation result from scanning mobility particle sizer (SMPS) is necessary to scale the measurements of SPAMS and HP-SPAMS to demonstrate the authentic size distributions of specific type particles. Because we didn't have SMPS to measure the ambient particles, size distribution of amine-containing particles was not scaled in this manuscript.
   "Generally, the particle size measured by HP-SPAMS ranges from 0.2 to 2.0 μm and was calibrated by polystyrene latex particles before and after the sampling campaign (Li et al., 2018; Li et al., 2011). The field observation result from scanning mobility particle sizer (SMPS) is necessary to scale the measurements of HP-SPAMS to demonstrate the authentic size distributions of specific type particles. Because the SMPS was not available in this sampling campaign, thus, the size distributions of amine-containing particles were not discussed here." has been added in the revised manuscript in the lines 199-206.

[Figure]

Figure R1. Hit rate and the number distribution of sized and detected ambient particles (Li et al., 2011).

**Specific comments and minor revisions:**

I might ask the authors to improve the English writing as below:

1. Line 58: "Further" should be "Furthermore".

    **Response:** "Further, greater than 80% of TMA- and DEA-containing particles internally mixed with nitrate, while the abundance of sulfate was higher in the DEA-containing particles (79.3%) than in the TMA-containing particles (55.3%)." have been revised to "Furthermore, greater than 80% of TMA- and DEA-containing particles internally mixed with nitrate, while the abundance of sulfate was higher in the DEA-containing particles (79.3%) than in the TMA-containing particles (55.3%)." in the lines 51-54.

2. Line 111-113: references should be cited here.

    **Response:** "In addition to the direct contribution of the SOA mass, the oxidation of amines by OH radicals, $NO_3$ radicals, and $O_3$ is also a substantial source of SOA production." have been revised to "In addition to the direct contribution of the SOA mass, the oxidation of amines by OH radicals, $NO_3$ radicals, and $O_3$ is also a substantial source of SOA production (Tong et al., 2020; Price et al., 2016)." in the lines 103-105.

3. Line 113: "($NO_3$ radicals, OH radicals, and ozone)" should be put in the right place.

**Response**: "Different amines (NO$_3$ radicals, OH radicals, and ozone) exhibit inconsistent behaviors under the same oxidation environments (Price et al., 2014; Silva et al., 2008; Murphy et al., 2007)." have been revised to "Different amines exhibit inconsistent behaviors under the same oxidation environments (NO$_3$ radicals, OH radicals, or ozone) (Silva et al., 2008; Price et al., 2014; Murphy et al., 2007)." in the lines 105-108.

4. Line 176: "three portions" should be "three parts".

**Response**: "The improvements and modifications from the SPAMS to the HP-SPAMS are comparatively presented below. The improvement in the SPAMS primarily includes three portions: the application of a concentration device, a delay extraction technology, and a multichannel acquisition technology (Chen et al., 2020; Li et al., 2018)." have been revised to "The improvements and modifications from the SPAMS to the HP-SPAMS are comparatively presented below. The improvement in the SPAMS primarily includes three parts: the application of a concentration device, a delay extraction technology, and a multichannel acquisition technology (Chen et al., 2020; Li et al., 2018)." in the lines 184-186.

5. Line 180: "delayed extraction technology" was not clearly described here, the authors should give more details about this technology.

**Response**: "Second, since the positions of the ionized ions scatter instead of being completely linear in the same direction, delayed extraction technology is used in SPAMS to replace the constant electrical field extraction technique. This delays the ions in order obtain sufficient potential energy in the appropriate time under a pulsed electric field and captures faster ions to improve the resolutions of positive and negative ions. The mass resolutions of the positive (> 1000 at maximum half width) and negative (> 2000 at maximum half width) ion spectra are then significantly improved." have been revised to "Second, the generated ions from the laser ionization of single particles firstly enter the zone without electric field. Then, the pulsed electric field will be added to accelerate the same kind of ions flying to the detector. This pulsed electric field instead of the constant electric field will prevent the initial deflection of same kind of ions, and the pulsed electric field also provides sufficient energy in the appropriate time to improve the resolutions of positive and negative ions. The mass resolutions of the positive (> 1000 at maximum half width) and negative (> 2000 at maximum half width) ion spectra are then significantly improved." in the lines 188-196.

6. L236, delete the should

**Response**: Revision made in the lines 258.

7. L240 Change "are presented in Figure 2" to (Figure 2)

**Response**: "The backward trajectories of the air masses (48 h, 500 m) associated with the spatial distributions of the two amine-containing particles during the entire sampling period are presented in Figure 2." have been revised to "The backward trajectories of the air masses (48 h, 500 m) associated with the spatial distributions of the two amine-containing particles during the entire sampling period (Figure 2)." in the lines 271-273.

8. L257 deleted the should. I might ask the authors carefully check the English. There are many "should" before the have been

**Response**: Thanks for your suggestion. We have revised "should" to "could" in the line 293. In addition, we have thoroughly checked the whole manuscript to avoid the same grammatical error.

9. L267, deleted the ambient

**Response**: Revision made in the line 303. "the particle counts of amine particles and the relative peak areas (RPAs) of amines in the particles with an increase in the ambient RH are presented in Figure 4" has been revised to "the particle counts of amine particles and the relative peak areas (RPAs) of amines in the particles with an increase in the RH are presented in Figure 4" in the lines 302-304.

10. Line 283: "Generally, TMA- and DEA-containing particles both contained amine marker ions, as well as organic fragments". The amine marker ions mentioned here were redundant since these single particles were selected based on the presence of amine markers.

**Response**: "Generally, TMA- and DEA-containing particles both contained amine marker ions, as well as organic fragments" have been revised to "Generally, TMA- and DEA-containing particles both contained organic fragments." in the lines 320-321.

11. Line 393: The upper axis in Figure 10 was not adequately presented.

**Response**: Revision made. The colors of lines are also revised.

The old Figure 10:

[Figure]

Figure 10. Temporal trends of TMA- and DEA-containing particle counts, and number fractions of nitrate, sulfate, and ammonium in TMA- and DEA-containing particles.

The new Figure 10:

[Figure]

Figure 10. Temporal trends of TMA- and DEA-containing particle counts, and number fractions of nitrate, sulfate, and ammonium in TMA- and DEA-containing particles.

12. Line 412: "the new Ra′ was reduced" should be "the new Ra′ reduced".

[revised manuscript text omitted]

---

## Author Comment (AC2)

**Response to the comments of Anonymous Referee #2**

[Atmospheric Chemistry and Physics, MS ID: acp-2021-593] Title: Diverse mixing states of amine-containing single particles in Nanjing, China

**General comments**

This article describes mixing state information for amine-containing particles derived exclusively using a single particle mass spectrometer at a sampling site in Nanjing. TMA and DEA were found to exhibit differing chemical mixing states, particularly with respect to oxidized organic and sulfate content. DEA-containing particles are proposed to be more acidic than TMA-containing particles based on the SPAMS peak area data for ammonium/aminium ions and sulfate/nitrate. The dependence of enhanced particulate TMA formation upon ambient RH has been demonstrated by others previously as cited in the article. Although not particularly novel, the article provides some new information on amine mixing state for this location. The text would benefit from a discussion of the drawbacks of single particle mass spectrometry when attempting quantitative or semiquantitative analyses (such as the acidity calculations). A discussion of the likely sources of gas phase TMA and DEA in the region would also be helpful.

**Response**: We are grateful to your comments on this paper. These valuable comments are of great help to the revision and improvement. We have addressed the comments and made substantial revisions based on studying these comments carefully. The responses and discussions about the drawbacks of single particle mass spectrometry and the possible sources of two amines are listed below. Our responses to the comments and revisions made in the manuscript are highlighted in blue color.

Anything about our paper, please feel free to contact me at <a href="mailto:chengel@jnu.edu.cn">chengel@jnu.edu.cn</a>

Best regards! Sincerely yours

Chunlei Cheng October 11, 2021

**Specific comments and point by point responses:**

**Major Comments:**

1. The text would benefit from a discussion of the drawbacks of single particle mass spectrometry when attempting quantitative or semiquantitative analyses (such as the acidity calculations).

**Response:** Indeed, it is difficult to accurately quantify the mass concentration of amines in the particles by HP-SPAMS due to the size-dependent transmission efficiencies of particles through aerodynamic lens and composition dependent matrix effect; thus, the quantitative relationship between two amines with sulfate, nitrate and ammonium (SNA) was not considered in this work. Currently, the results were used to illustrate the different effects of same influencing factors on the behaviors of amine-containing particles based on their mixing states.

The relative acidity ratio used in this work is not an equivalent result of the calculation of mass concentrations of anions and cations. It is roughly estimated from the peak areas of SNA, which is intend to compare the relative particle acidity of different amine-containing particles. This parameter is only referred in the studies from single particle mass spectrometry. Huang et al. (2013) have compared the actual particle acidity calculated from inorganic ions (MARGA data) and relative acidity ratio obtained from single particle mass spectrometer (ATOFMS) (Huang et al., 2013). The comparison result is as follows:

This graph is from the field study of Huang et al. (2013). The robust linear regression between ATOFMS particle acidity and MARGA particle acidity was obtained, which indicates a feasible estimation of particle acidity through the peak areas of sulfate, nitrate and ammonium obtained from single particle mass spectrometry.

"It should be noted that HP-SPAMS measurements cannot provide the quantitative mass concentrations of amines and related chemical species due to the

size-dependent transmission efficiencies of particles through aerodynamic lens and composition dependent matrix effect (Cheng et al., 2018; Cheng et al., 2021; Gong et al., 2021). Currently, the observational results were used to illustrate the distinct impacts of same influencing factors on the behaviors of amine-containing particles." has been added in the lines 229-235.

"Although the feasibility of Ra has been supported by the robust linear correlation with authentic particle acidity calculated from mass concentrations of inorganic ions (Huang et al., 2013), this semi-quantitative approach should be carefully treated when it comes to the discussion about the actual acidity of atmospheric particles." has been added in the lines 444-448.

2. A discussion of the likely sources of gas phase TMA and DEA in the region would also be helpful.

**Response:** According to reported studies the gaseous TMA and DEA are mainly from agriculture, industry, vehicle exhaust, biomass combustion, biological, and marine sources (Zhou et al., 2019; Hemmiläet al., 2018; Sintermann et al., 2014; Ge et al., 2011; Zhang et al., 2017). Their concentrations vary greatly depending on the influence of source strength near the sampling site. Since HP-SPAMS measurements cannot provide the quantitative mass concentrations of amines, thus, it is difficult to resolve the proportions of gaseous TMA and DEA from their major sources. We have added some discussions about the gaseous sources of TMA and DEA, which will help to indicate their potential sources in this work.

"The gaseous TMA and DEA are mainly from agriculture, industry, vehicle exhaust, biomass combustion, biological, and marine sources (Zhou et al., 2019; Hemmiläet al., 2018; Sintermann et al., 2014; Ge et al., 2011; Zhang et al., 2017). Their concentrations vary greatly depending on the influence of source strength near the sampling site. For example, the gaseous concentration of DEA was 14 and 2-5 times higher than TMA in polluted urban areas in China (Yao et al., 2016) and US (You et al., 2014), respectively, while higher concentration of TMA than DEA was observed in the forest site (You et al., 2014). Both the online and offline measurements are difficult to quantitatively resolve their emission sources (You et al., 2014; Yao et al., 2016; Kieloaho et al., 2013; Hellén et al., 2014). Here the backward trajectories of the air masses from sampling site were discussed to explorer their possible different sources. "has been added in the lines 260-271.

**3. Do the back trajectories help to inform the dominant sources of TMA vs DEA?**

**Response:** As we have discussed in the original manuscript based on the Figure 2, TMA-containing particles were primarily from the air masses of Cluster 1 and Cluster 4, while the DEA-containing particles were associated with the air masses of Cluster 3 and Cluster 4, which underwent long-range transport. The Cluster 1, 3 and 4 were all associated with the strong anthropogenic sources in the YRD (eastern of the sampling site) and marine sources in the East China Sea. Unfortunately, the rough spatial resolution of backward trajectories of the air masses was incapable in matching

the real-time changes of amine-containing particles detected by HP-SPAMS. Thus we cannot locate the exact anthropogenic source for TMA and DEA via the results of backward trajectories. Therefore, we kept this part unchanged and no further discussions were added in the revision stage.